# Influence of Living Arrangements and Eating Behavior on the Risk of Metabolic Syndrome: A National Cross-Sectional Study in South Korea

**DOI:** 10.3390/ijerph16060919

**Published:** 2019-03-14

**Authors:** Heesook Son, Hyerang Kim

**Affiliations:** Red Cross College of Nursing, Chung-Ang University, 84 Heukseok-ro, Dongjak-gu, Seoul 06974, Korea; hson@cau.ac.kr

**Keywords:** eating behavior, living arrangement, metabolic syndrome, dietary patterns, adults

## Abstract

Studies on the relationships between health, different living arrangements, and eating behaviors across age groups are limited. Therefore, we investigated these associations, focusing on metabolic syndrome, among 16,015 South Koreans aged ≥19 years who completed the Korean National Health and Nutrition Examination Survey (2013–2016). Multivariate logistic regression revealed that younger adults (<65 years) who lived and ate alone consumed more carbohydrates than those who lived and ate with others (*p* < 0.01). The odds of metabolic syndrome in younger adults increased with eating alone (adjusted odds ratio (aOR) 2.11, 95% confidence interval (CI) 1.10–4.02) and living and eating alone (2.39, 1.25–4.58). Older adults (≥65 years) did not differ in dietary intake or prevalence of metabolic syndrome according to their living and eating situations. Younger adults living and eating alone may benefit from customized nutrition and health management programs to reduce their risk of metabolic syndrome.

## 1. Introduction

Metabolic syndrome (MetS) is associated with a substantially greater risk of all-cause mortality. With a worldwide prevalence in the adult population of approximately 20–30%, MetS has noticeably aggravated the global health burden [1]. Unhealthy dietary patterns, including meal irregularity, skipping meals, and eating out, are known to play a critical role in the development of MetS [2]

Changing economic, cultural, and social conditions have contributed to an increase in the number of one-person households [3]. In 2016, 13% of all households worldwide and 27.9% of all households in Korea were one-person households [4]. Living alone is associated with a number of negative health outcomes, such as diabetes, cardiovascular disease, and obesity [5,6]. These findings suggest that living alone possibly influences dietary intake and eating behaviors [7]. Indeed, as society shifts toward a more individualized, informal, and atypical eating style, an increasing number of individuals appear to be eating alone [8]. Eating behavior, including eating frequency and duration, eating out, meal companionship, and meal types, influences food intake and nutritional status [8]. Eating alone specifically has a negative influence on dietary patterns and diet quality and may lead to nutritional inadequacy and imbalance as well as undesirable dietary behaviors known to be risk factors for MetS [9]. However, the interactions between eating behavior, dietary choices, and food intake and their impact on nutritional well-being and long-term health remain unclear.

Younger adults more frequently experience changes in living arrangements and lifestyle (i.e., for their job) than older adults, making the dietary patterns of younger adults more susceptible to change [10]. Furthermore, their more active social lives restrict their time for at-home meal preparation, making them more likely to eat out [11]. Young people, in leaving their parents and living independently, experience numerous diet-related behavioral changes, some of them are unfavorable (e.g., frequent exposure to energy-dense nutrient-poor foods, lower fruit and vegetable intake, frequently skipping meals [12]).

The rapidly aging population and increased number of older adults living alone in South Korea has raised concerns in that such older adults are more vulnerable to unhealthy dietary behaviors and poor health outcomes [13]. A review showed that older adults living alone experience economic strain and psychosocial disadvantages (e.g., cognitive impairment, social isolation), and struggle in daily living, which can contribute to reduced food intake, dietary imbalance, poor nutritional status, and a high risk of adverse health outcomes [7]. Therefore, greater attention must be paid to the influence of social interactions in specific living arrangements and eating behavior on dietary intake, nutritional status, and health outcomes in older adults.

Living arrangements and eating behavior have been explored in relation to various health outcomes [5]. However, these studies were limited to specific groups of children, adolescents, or older adults—they did not consider several age groups at once, despite clear differences in lifestyles and dietary patterns between these groups. Furthermore, the bivariate relationships of these variables were explored, rather than different combinations of living arrangements and eating behaviors [9,13]. Therefore, in the current study, we compared the prevalence of different combinations of living arrangements and eating behaviors (focusing on living and/or eating alone) between younger (<65 years) and older adults (≥65 years) in a representative sample of Koreans. We also investigated the association of these combinations with dietary intake and the risk of MetS, and how the associations differed by age.

## 2. Materials and Methods

### 2.1. Study Population

This secondary data analysis used data from the sixth wave (2013–2016) of the Korean National Health and Nutrition Examination Survey (KNHANES). The KNHANES offers a broad perspective on health risks and nutritional status via health and dietary interviews, standardized physical examinations, and laboratory tests [14]. The inclusion criteria were being 19 years or older and providing information about nutritional intake in the 24-h recall interview. To avoid measurement bias, respondents who consumed less than 500 or more than 5000 kcal/day were excluded. Of the 18,034 eligible participants, 16,015 were ultimately analyzed (5387 in 2013, 5322 in 2014, and 5306 in 2015). The institutional review board where the authors affiliated approved this current study for secondary data analysis (IRB No. 1041078-201712-HRBM-240-01).

### 2.2. Metabolic Syndrome

Using the National Cholesterol Education Program’s Adult Treatment Panel III criteria for Asians [15], we defined MetS as satisfying at least three of the following criteria: (1) waist circumference ≥90 cm for men and ≥80 cm for women; (2) triglycerides ≥150 mg/dL; (3) high-density lipoprotein (HDL) cholesterol <40 mg/dL for men and <50 mg/dL for women; (4) systolic blood pressure ≥130 or diastolic blood pressure ≥85 mm Hg; and (5) fasting blood glucose ≥100 mg/dL.

### 2.3. Living Arrangements and Eating Behavior

Whether respondents were living alone was assessed by asking “How many family members are living with you?” We took a response of “zero” to mean that the respondent was living alone. Eating alone was assessed using the question “Did you usually have [breakfast, lunch, or dinner] with others in the past year?” If the response for each meal was “no,” the participant was categorized as “exclusively eating alone”; if a participant made at least one “yes” response, they were categorized as “eating with others.” We created four groups based on living arrangements and eating behavior: “living with others and eating with others” (LW × EW, *n* = 13,864), “living with others and exclusively eating alone” (LW × EA, *n* = 646), “living alone and eating with others” (LA × EW, *n* = 774), and “living alone and exclusively eating alone” (LA × EA, *n* = 731).

### 2.4. Dietary Intake

Dietary intake was assessed via the 24-h recall method. Seven dietary factors were evaluated: total energy intake (kcal/day), total carbohydrate intake (g/day), total protein intake (g/day), total fat intake (g/day), and percentage of energy derived from each of carbohydrates, protein, and fat. Contributions of macronutrients to total energy consumption were compared to the acceptable macronutrient distribution ranges (AMDRs) for carbohydrates, protein, and fat. According to the 2015 dietary reference intakes for Koreans, the AMDRs for Koreans are 55–65% for carbohydrates, 7–20% for protein, and 15–30% for fat [16].

### 2.5. Covariates

To examine the independent associations of living arrangements and eating behavior with the outcome variable, known risk factors for MetS were assessed as covariates. The sociodemographic variables consisted of age, sex, household income, education, and employment status. Health-related variables included smoking status (never smoked, past smoker, or current smoker), alcohol consumption (lifelong experience of alcohol consumption or not), and walking hours per week (as an index of physical activity).

### 2.6. Statistical Analysis

The KNHANES used a multistage stratified cluster sampling method [17]. Accordingly, we analyzed the data using a weighting methodology and accounted for masked variance, as proposed in the guidelines of the Korean Centers for Disease Control and Prevention for the analysis of complex KNHANES data [14].

The chi-square test was used to determine differences in categorical variables (socio-demographic, health-related, dietary behavior characteristics) between the living arrangements and eating behavior groups. One-way analysis of variance (ANOVA) was used to compare the means of the dietary intake factors and risk factors for MetS among the living arrangements/eating behavior groups. Post-hoc comparisons were made using Tukey’s test.

To examine the associations of living arrangements and eating behavior with MetS in each age group, logistic regression analysis was conducted. Adjusted odds ratios (aORs) with 95% confidence intervals (CIs) were estimated while adjusting for age, sex, income, education, smoking, alcohol consumption, physical activity, and total energy intake. A *p*-value < 0.05 indicated statistical significance. All statistical analyses were performed using IBM SPSS Statistics 23.0 (New York, NY, USA).

## 3. Results

We observed a significant difference in dietary intake by living arrangements and eating behavior only among younger adults (Table 1); namely, the LA × EA group consumed the highest amount of carbohydrates (*p* < 0.01) and obtained more of their energy from carbohydrates (*p* < 0.01), such that their intake exceeded the carbohydrate AMDR of 55–65%. Among older adults, none of the dietary factors differed significantly by living arrangements and eating behavior. However, for all groups, the percentage of energy consumed from carbohydrates and fat exceeded the corresponding AMDRs (over 65% for carbohydrates and lower than 15% for lipids).

We observed differences in risk factors for MetS by living arrangements and eating behavior only among younger adults (Table 2 and Appendix A): the number of risk factors for MetS was highest in the LA × EA group (*p* < 0.05). They also had the highest prevalence of abdominal obesity, high blood pressure, and high fasting blood glucose (*p* < 0.01). For older adults, the prevalence of abdominal obesity and low HDL cholesterol was higher in the living alone groups compared to the living with other groups (*p* < 0.01), regardless of eating behavior. The prevalence of MetS was highest in younger adults in the LA × EA group and lowest among older adults in the LW × EW group (*p* < 0.01). Table 3 shows the regression analysis results. Compared to the LW × EW group, living alone and eating alone independently increased the odds of high fasting blood glucose in younger adults (adjusted odds ratio (aOR) of 1.22 with 95% confidence interval (CI) of 0.75–1.99 for LA × EW and aOR of 1.98 with 95% CI of 1.04–3.75 for LW × EA); the odds were even higher in the LA × EA group (aOR 2.85, 95% CI 1.41–5.77). The odds of MetS were also higher in the LW × EA group (aOR 2.11, 95% CI 1.10–0.02)) and the LA × EA group (aOR 2.39, 95% CI 1.25–4.58) (*p* < 0.05). We found no significant relationship between MetS among older adults and living arrangements or eating behavior.

## 4. Discussion

While there are noticeable lifestyle differences between younger and older adults, it is unclear if the associations of living arrangements and eating behavior with dietary intake and health outcomes differ by age. We therefore examined the associations between different combinations of living arrangements and eating behaviors with dietary intake and risk of MetS in younger and older Korean adults. There were three major findings.

First, living and eating alone was significantly related to an unbalanced dietary intake and the risk of MetS. This is consistent with previous studies indicating that living and eating alone are associated with poor dietary quality [9]. Living alone (vs. living with family) is associated with greater consumption of food high in carbohydrates and fat [9] and eating alone with greater intake of dietary energy and carbohydrates, and less food diversity [18]. A high-carbohydrate diet makes a sizeable contribution to the prevalence of MetS [17]. The traditional rice-oriented diet in Korea (characterized by a staple diet of white rice) is a major reason for the high carbohydrate consumption among Koreans and is significantly associated with MetS and cardiovascular disease [19]. Thus, the high carbohydrate intake of individuals who lived and ate alone could explain their high MetS risk.

People living and eating alone also had a poorer socioeconomic status (Appendix A), which possibly explains their higher risk of MetS. Socioeconomic constraints are linked to unhealthy dietary intake and behaviors and worse health management, which are strong contributors to MetS [20]. Previous studies supported that being male, older, unmarried, in a smaller household, of lower income, and of urban residence are closely related to eating alone, while individuals with low socioeconomic status consume more carbohydrates and less protein and fat [21].

This first result suggests that in order to identify dietary problems and the risk factors for MetS it is necessary to contextualize dietary characteristics within various eating situations. Future studies should explore the mechanisms of how dietary behaviors relate to MetS, such as by including more detailed information on those behaviors (e.g., frequency of eating alone and eating out, reasons for doing so, and type of meals).

Second, the associations of living arrangements and eating behavior with dietary patterns and MetS were found only in younger adults. This suggests that younger adults’ dietary patterns and health are more sensitive to the effects of living and eating alone. The dietary habits of younger adults are highly changeable, possibly due to their more frequent experience of life changes (e.g., independence from parents, relocation to new physical and social environments [22]). A lack of cooking skills and nutritional knowledge might also prevent younger adults from preparing meals at home, prompting them to eat out [23]. They also tend to perceive greater time scarcity, preferring to adopt more time-saving dietary behavior such as eating out, skipping meals, or consuming fast food [24]. When they do eat at home, they might consume snacks or convenience foods, which increases their carbohydrate consumption and lowers the food diversity of their meals [24,25].

Given our results, the socialization of eating might be important for improving dietary quality and health outcomes in younger adults. Individuals’ social connectedness and relationships might contribute to their frequency of eating out: individuals with meal companions—even ones who live separately—or who are more engaged in social activities tend to have more opportunities to eat meals with others and eat out, which can potentially improve their dietary quality and variety [25]. However, eating out can also lead to dietary imbalances, increasing total energy, fat, and sugar intake while lowering fiber, calcium, and vitamin C intake [26]. The result suggests that interventions may be necessary for younger adults to support them in making healthier food choices when eating out and to help them develop their cooking skills for meal preparation at home.

Finally, older adults’ macronutrients intake failed to adhere to the AMDRs in all living and eating situations, such that they consumed a higher proportion of carbohydrates and a lower proportion of lipids than recommended. This may be because older Korean adults tend to adhere more to the traditional Korean dietary pattern as they age, which is characterized by higher carbohydrate intake [27]. Our finding that dietary intake did not differ by living arrangements and eating behaviors among older adults contradicts previous findings that living alone is detrimental to dietary and health outcomes in older adults. Elderly individuals living alone have been characterized as consuming a less diverse diet, and also characterized by a lower intake of fruits, vegetables, and fish and a higher intake of carbohydrates and salty foods [28]. Furthermore, previous reports suggested that individuals eating alone have lower dietary quality overall, eating less meat and vegetables and consuming more calories and carbohydrates [29]. Further studies are necessary to investigate the physical and psychosocial environments of older adults in more detail (e.g., family structures, housemate types, social networks and relationships, meal companionship) in order to explain the links of living arrangements and/or eating behavior with dietary intake and health outcomes in this age group. This would help address the challenges associated with an increasing population of individuals living and eating alone.

### Limitations

There are some limitations to this study. First, we divided eating behavior into two categories only: eating with others and exclusively eating alone. A previous study reported a dose–response relationship between health outcomes and frequency and duration of eating alone [25,30]. Future studies should therefore include the frequency and duration of eating alone to provide more nuanced results. Second, we obtained no information about meal companionship when respondents ate with others. Meal companionship influences the amount and type of food intake [31]. Examining more detailed eating situations, including the reasons for eating alone, meal companionship, and quality of food (e.g., meal at home, ready-to-eat meal, restaurant meal), would help us better understand the association between dietary patterns and health. Third, we had limited information on the duration of living and eating alone and family structure. Lifestyle and life skills might differ according to the duration of certain living arrangements and eating behaviors, while eating patterns might vary with family composition and heterogeneity of family members living together (e.g., their gender and age). Thus, these factors should be explored in future studies. Fourth, the relationships between psychological aspects and living and/or eating conditions were not included in this analysis, which were examined in previous studies. Particularly in older people, for example, eating alone may increase depression, while depression could be also aggravated by living alone [32,33]. In addition, diet quality and meal frequency were negatively associated with depression [34]. To determine the bilateral relationship between depression and metabolic syndrome [35], further study is needed to identify the links between psychological aspects, eating behaviors, and living conditions. Finally, since this was a cross-sectional study, we cannot infer causality of the relationships. A prospective design is needed to investigate the effect of living and eating alone on the development of MetS.

## 5. Conclusions

We found that living arrangements and eating behavior had age-dependent associations with diet and the risk of MetS, with only younger adults showing significant relationships. These results call for customized interventions for young adults with risky living arrangements and eating behaviors. For example, a self-management support program for life skill development might help younger people living and eating alone learn more about their dietary behavior and take a proactive role in forming healthy dietary patterns. In the future, we suggest examining how other variances in contemporary socioenvironmental contexts influence health-related lifestyle and health outcomes.

## Figures and Tables

**Table 1 ijerph-16-00919-t001:** Nutrient intake by living arrangements and eating behavior for each age group ^1.^

	<65 years				*p*	≥65 years				*p*
	LW × EW ^2^ (*n* = 10,765)	LW × EA (*n* = 363)	LA × EW (*n* = 467)	LA × EA (*n* = 165)		LW × EW (*n* = 3099)	LW × EA (*n* = 283)	LA × EW (*n* = 307)	LA × EA (*n* = 566)	
Total energy (kcal)	1889.5 ± 45.9	1853.0 ± 103.5	1899.97 ± 81.35	1990.7 ± 150.7	0.90	1580.2 ± 66.7	1523.3 ± 93.2	1648.9 ± 104.9	1704.6 ± 94.5	0.30
Carbohydrate (g)	346.6 ± 5.0	367.7 ± 1.0	331.9 ± 9.3	371.9 ± 12.8	<0.01	328.0 ± 5.0	319.0 ± 9.6	308.7 ± 15.0	324.5 ± 7.8	0.41
Protein (g)	64.7 ± 1.6	63.46 ± 3.6	63.2 ± 2.7	63.7 ± 4.8	0.20	51.0 ± 2.2	48.4 ± 3.0	50.7 ± 3.2	55.0 ± 3.1	0.91
Lipids (g)	54.79 ± 1.5	54.71 ± 3.4	58.5 ± 2.4	55.3 ± 3.8	0.43	33.4 ± 14.4	33.3 ± 2.3	35.2 ± 4.1	34.2 ± 2.0	0.96
% Energy from carbohydrate	64.9	67.3	61.8	69.1	<0.01	71.6	70.3	68.9	70.5	0.56
% Energy from protein	13.7	13.7	13.3	12.8	0.10	12.9	12.7	12.3	12.9	0.72
% Energy from fat	18.9	18.6	20.1	17.8	0.23	13.9	13.8	14.4	14.1	0.99

^1^ Adjusted for age, sex, income, education, smoking, alcohol consumption, and physical activity. ^2^ LW × EW: Living with others and eating with others; LW × EA: Living with others and eating alone; LA × EW: Living alone and eating with others; LA × EA: Living alone and eating alone.

**Table 2 ijerph-16-00919-t002:** Risk factors for metabolic syndrome for each age group ^1^.

	<65 years				*p*	≥65 years				*p*
	LW × EW ^2^ (*n* = 10,765)	LW × EA (*n* = 363)	LA × EW (*n* = 467)	LA × EA (*n* = 165)		LW × EW (*n* = 3099)	LW × EA (*n* = 283)	LA × EW (*n* = 307)	LA × EA (*n* = 566)	
Waist circumference (male ≥ 90 cm, female ≥ 80 cm)	3128 (31.0)	139 (36.4)	163 (30.5)	77 (45.9)	<0.01	1349 (49.2)	148 (53.6)	183 (62.0)	331 (61.4)	< 0.01
Triglycerides (≥150 mg/dL)	989 (15.2)	38 (14.3)	67 (19.3)	28 (23.8)	0.05	280 (7.1)	28 (9.0)	33 (6.3)	46 (9.6)	0.70
HDL cholesterol (male < 40, female < 50 mg/dL)	2899 (29.5)	107 (27.8)	136 (29.0)	64 (40.6)	0.16	997 (42.9)	91 (47.4)	118 (51.1)	202 (51.8)	0.01
Blood pressure (≥130/≥85 mmHg)	1763 (17.6)	77 (21.1)	119 (21.0)	60 (32.4)	<0.01	1578 (60.1)	154 (63.7)	185 (61.3)	328 (63.8)	0.45
Fasting blood glucose (≥110 mg/dL)	620 (6.4)	43 (13.3)	32 (6.5)	36 (23.2)	<0.01	512 (23.6)	49 (30.8)	50 (21.3)	90 (23.9)	0.17
Metabolic syndrome (three or more risk factors)	778 (11.1)	44 (15.8)	50 (12.1)	38 (30.4)	<0.01	558 (30.1)	58 (39.6)	70 (38.1)	115 (37.8)	<0.01

^1^ Adjusted for age, sex, income, education, smoking, alcohol consumption, physical activity, and total energy intake. ^2^ LW × EW: Living with others and eating with others; LW × EA: Living with others and eating alone; LA × EW: Living alone and eating with others; LA × EA: Living alone and eating alone; MetS: metabolic syndrome; HDL: high-density lipoprotein.

**Table 3 ijerph-16-00919-t003:** Adjusted odds ratios of the relationship of living arrangements and eating behavior with risk factors for metabolic syndrome ^1^.

	<65 years				*p*	≥65 years				*p*
	LW × EW ^2^ (*n* = 10,765)	LW × EA (*n* = 363)	LA × EW (*n* = 467)	LA × EA (*n* = 165)		LW × EW (*n* = 3099)	LW × EA (*n* = 283)	LA × EW (*n* = 307)	LA × EA (*n* = 566)	
Waist circumference (male ≥ 90 cm, female ≥ 80 cm)	ref	1.12 *	1.37	1.58	0.22	ref	1.28	0.87	1.19	0.65
		(0.77–1.65)	(0.83–2.25)	(0.90–2.78)			(0.74–2.21)	(0.49–1.55)	(0.77–1.84)	
Triglycerides (≥150 mg/dL)	ref	1.15	1.13	1.23	0.84	ref	0.86	1.16	1.07	0.96
		(0.77–1.71)	(0.62–2.07)	(0.62–2.41)			(0.39–1.88)	(0.46–2.92)	(0.53–2.14)	
HDL cholesterol (male < 40, female < 50 mg/dL)	ref	1.02	0.83	1.17	0.85	ref	0.88	0.87	0.82	0.87
		(0.71–1.48)	(0.48–1.44)	(0.69–1.99)			(0.48–1.60)	(0.44–1.72)	(0.49–1.39)	
Blood pressure (≥130/≥85 mmHg)	ref	1.24	1.34	1.41	0.28	ref	1.11	1.21	1.16	0.84
		(0.89–1.72)	(0.72–2.49)	(0.82–2.40)			(0.64–1.91)	(0.69–2.12)	(0.72–1.87)	
Fasting blood glucose (≥110 mg/dL)	ref	1.22	1.98	2.85	<0.01	ref	0.93	1.13	0.77	0.85
		(0.75–1.99)	(1.04–3.75)	(1.41–5.77)			(0.49–1.78)	(0.57–2.21)	(0.41–1.48)	
Metabolic syndrome (three or more risk factors)	ref	0.92	2.11	2.39	0.01	ref	1.22	1.06	0.85	0.86
		(0.56–1.50)	(1.10–4.02)	(1.25–4.58)			(0.60–2.46)	(0.48–2.39)	(0.45–1.63)	

* Adjusted odds ratio (95% confidence interval) ^1^ Adjusted for age, sex, income, education, smoking, alcohol consumption, physical activity, and total energy intake. ^2^ LW × EW: Living with others and eating with others; LW × EA: Living with others and eating alone; LA × EW: Living alone and eating with others; LA × EA: Living alone and eating alone.

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
