# Peer review of "Influence of Living Arrangements and Eating Behavior on the Risk of Metabolic Syndrome: A National Cross-Sectional Study in South Korea"

_ijerph, 2019, doi:10.3390/ijerph16060919_

Round 1

Reviewer 1 Report

Comments to the Authors

The topic of potential influence of eating habits and living arrangements on the risk of Metabolic Syndrome at Koreans of all ages is interesting, but in the current version, the text has some flaws.

1. The Authors  prepared this manuscript with great care. Statistical calculations are made at a high level. Unfortunately, the presentation of the results is not so well-designed. The tables are too large, contain too many details.

- first of all tables should be smaller, which may result in their subdivision. Especially Tables 1 and 3 are not very legible and the data contained in them should be placed in additional tables.

 - in Table 1, the Authors of the manuscript presented data with an accuracy of one decimal place. I would suggest the consequences in the presentation of data in Tables 2 and 3. The more so that the presentation of data to one decimal place will affect the readability of the tables.

- Also in table 4, especially that a 95% confidence interval is given, data should only be provided to the decimal place only.

2.  In the discussion, the Authors of manuscript could write about the relationship between the inferior mental condition of lonely people (depressive states) with eating disorders and lifestyle, especially of older people. Poor mental health can indirectly increase the risk of MS

With best regards!

Author Response

Point 1: The topic of potential influence of eating habits and living arrangements on the risk of Metabolic Syndrome at Koreans of all ages is interesting, but in the current version, the text has some flaws.

 The Authors prepared this manuscript with great care. Statistical calculations are made at a high level. Unfortunately, the presentation of the results is not so well-designed. The tables are too large, contain too many details.

- first of all tables should be smaller, which may result in their subdivision. Especially Tables 1 and 3 are not very legible and the data contained in them should be placed in additional tables.

 - in Table 1, the Authors of the manuscript presented data with an accuracy of one decimal place. I would suggest the consequences in the presentation of data in Tables 2 and 3. The more so that the presentation of data to one decimal place will affect the readability of the tables.

- Also in table 4, especially that a 95% confidence interval is given, data should only be provided to the decimal place only.

Response 1: Thank you for your positive feedback. As you indicated, we revised the tables to improve readability. We replaced the two tables of 1 and 2 to Supplementary tables. We sent the Table 1 to “Supplementary table 1” We definitely agree with you that the table 3 looks confusing. We modified the table 3 to improve readability. The descriptive statistics of measure of risk factors was sent to “Supplementary table 2” to present the main result in the table. Finally, all table numbers were accordingly changed through the manuscript. We have also corrected decimal places. For table 3 that was table 4 before this revision, however, we thought that two decimals would be better to be presented for a 95% confidence interval. We have read other articles published in this journal and found two decimals are presented in tables. Therefore, we have left two decimals for 95% confidence interval at this point. Your advice would be appreciated.

Point 2: In the discussion, the Authors of manuscript could write about the relationship between the inferior mental condition of lonely people (depressive states) with eating disorders and lifestyle, especially of older people. Poor mental health can indirectly increase the risk of MS

Response 2: Thank you for pointing this out. We definitely agree with you that it is needed to include the relationship between metal condition, eating disorder and lifestyle for old adults. To reflect the comment, we have included the sentences in the section of limitations.

Reviewer 2 Report

Reviewer’s Comments:

Overall, this is quite an interesting and a strong manuscript. I think the introduction is well written. Authors did a good job on the methods, it is detailed and concise.  The data analysis is strong. Results is thorough and well presented. The discussion and conclusion are pretty robust.

But I was wondering how authors obtained the informed consent from participants. On line 68, authors indicated that they used data from their 2013-2016 national health and nutrition survey for this paper. This means that they used existing or secondary data to prepare the manuscript. My question is, how did they obtain the informed consents from all participants included in the study?  This is stated on lines 76-77). Are they making reference to the informed consent that was sought from participants when the original data was collected? Or authors reached out to all participants included in this "secondary data analysis" to seek informed consent again before they used the secondary data? This should be clarified.

Author Response

Point 1: Overall, this is quite an interesting and a strong manuscript. I think the introduction is well written. Authors did a good job on the methods, it is detailed and concise.  The data analysis is strong. Results is thorough and well presented. The discussion and conclusion are pretty robust.

But I was wondering how authors obtained the informed consent from participants. On line 68, authors indicated that they used data from their 2013-2016 national health and nutrition survey for this paper. This means that they used existing or secondary data to prepare the manuscript. My question is, how did they obtain the informed consents from all participants included in the study?  This is stated on lines 76-77). Are they making reference to the informed consent that was sought from participants when the original data was collected? Or authors reached out to all participants included in this "secondary data analysis" to seek informed consent again before they used the secondary data? This should be clarified.

Response 1: Thank you for all of your positive comments and feedback. For this study, secondary data analysis was utilized from the national dataset, Korean National Health and Nutrition Examination Survey (KNHANES) that was conducted nationwide based on standardized protocol. We definitely agree with you that the sentence of “informed consent was obtained from all participants included in the study” is confusing. Therefore, the sentence was removed. The data file of KNHANES is open to public use since it is the national survey, but we obtained IRB for secondary data analysis. We have revised the section of study population including the reference of KNHANES. 

Round 2

Reviewer 1 Report

The manuscript has been improved as expected and does not require further adjustment.